
# Status and future prospects of the NEWS-G experiment

P. Knights$^\star$ on behalf of the NEWS-G collaboration

School of Physics and Astronomy, University of Birmingham, B15 2TT, UK

$\star$ p.r.knights@bham.ac.uk

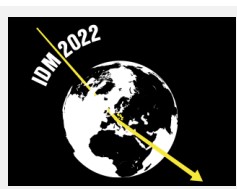

## Abstract

The NEWS-G collaboration is searching for light dark matter using spherical proportional counters. Access to the mass range from 50 MeV to 10 GeV is enabled by the combination of low energy threshold, light gaseous targets (H, He, Ne), and highly radio-pure detector construction. Several of the recent developments that have paved the way for the operation of a new 140 cm in diameter, spherical proportional counter are presented. Constructed and commissioned at LSM using 4N copper with a $500\,\mu x\,m$ electroplated inner layer, the detector is now installed in SNOLAB, where it has begun data taking. Building on the electroplating of the current detector, the collaboration construct its next detector directly in the underground laboratory, using ultra-pure copper electroforming. The design and construction of ECUME, a 140 cm in diameter spherical proportional counter fully electroformed underground is discussed, along with the potential to achieve sensitivity reaching the neutrino floor in light Dark Matter searches with a next-generation detector.

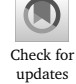
## 1 Introduction and status

Direct searches for Dark Matter (DM) have been carried out for several decades, primarily focussed in the mass range $10-1000$ GeV, with a recent review presented in Ref. [1]. However, theoretical and experimental interest has recently been attracted to the light-DM region, below a few GeV, which remains relatively unexplored [2–4]. Access to this region of parameter space requires either novel detection channels [5] or lower energy thresholds and light targets [1].

The New Experiments With Spheres - Gas (NEWS-G) collaboration is searching for light-DM using an innovative gaseous detector, the spherical proportional counter [6–9], shown in Figure 1. First results with a $\varnothing$60cm prototype detector operated in the Modane Underground Laboratory (LSM), France, were obtained with a 9.7 kg·day exposure with a Ne:CH$_4$ gas

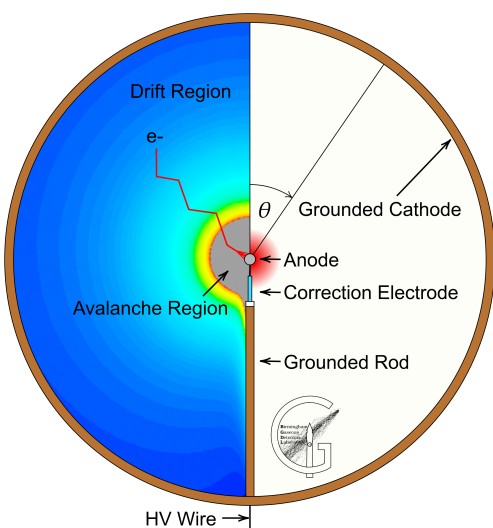

Figure 1: Schematic and principle of operation of the spherical proportional counter [6].

mixture. This produced the most stringent constraint at the time on the spin-indpendent DM-nucleon scattering cross-section, $4.4 \times 10^{-37}$ cm$^2$, for a 0.5 GeV mass DM candidate [10]. This data have also recently been used to perform a search for Kaluza-Klein axions [11].

NEWS-G have since constructed a $\varnothing$140 cm detector [12]. Shown in Figure 2, the detector benefits from increased radiopurity material construction, namely 99.99% pure copper with an ultra-pure copper electroformed internal layer [13], and an improved, compact shielding.

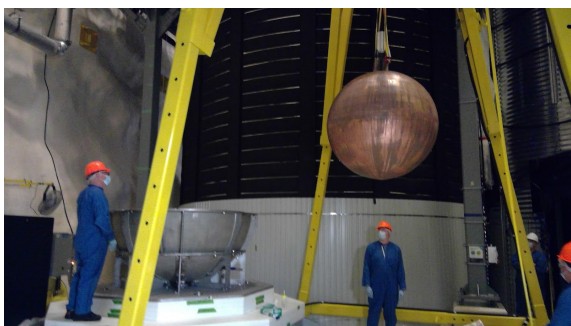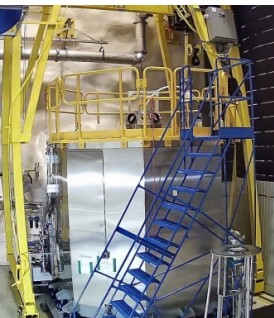

Figure 2: Left: The 140 cm in diameter NEWS-G spherical proportional counter being installed in SNOLAB. Right: The detector's shielding. The shield is comprises 3 cm archeological lead, 22 cm low activity lead, and 40 cm high-density polyethylene, encased in a stainless steel skin.

The detector was constructed and commissioned in LSM. Commissioning data was collected with the detector installed in a temporary shielding. Analysis of approximately 10 days of data collected with the detector filled with 135 mbar CH$_4$ is currently ongoing, with an update reported in Ref. [14]. Following this commissioning phase, the detector was shipped to SNOLAB, Canada. NEWS-G have recently begun collecting physics-quality data with the detector installed inside its compact shielding [14].

To support current and future DM searches, several developments have been necessary. In order to efficiently collect electrons generated in the 140 cm detector and improve detector

stability, a multi-anode read-out sensor, ACHINOS, has been developed [15–17]. Gas purification methods to remove contaminant gases, such as oxygen, water and radon, have also been studied, as have the production of purifiers with reduced radon emanation [18,19]. Furthermore, several measurements and estimations of the ionisation quenching factor have recently been performed, using various methods [20–22]. Further campaigns to extend measurements to even lower energies and perform measurements in other relevant gases are planned in the near future.

## 2  Future fully electroformed detectors

The current NEWS-G detector's dominant background is the radioactivity contamination of the 99.99% pure copper used for the vessel, which is either inherent to the material or cosmogenically induced during it's transportation [12]. The next generation NEWS-G detectors will be constructed using ultra-pure copper electroforming [23], with the detector being electroformed intact - removing the need for welding, which may introduce additional background - and directly in the underground lab where it will be hosted, to minimise cosmogenic activation. This is the motivation behind the ECuME project, which will deliver a 140 cm in diameter detector, to be installed into the current detector's shielding upon completion of its physics exploitation stage. A 30 cm in diameter prototype detector will being construction late 2022, and will define the electroforming parameters for ECuME.

A future 3 m in diameter fully electroformed detector, DarkSPHERE, is currently being designed. DarkSPHERE will employ a water-based shielding, to suppress background contributions from the shielding material. The Boulby Underground Laboratory, UK, is being targeted as a host, with the available space in the Large Experiment Cavern sufficient to house $3 \times 3 \times 3$ m$^3$ shield, which would enable DarkSPHERE to achieve a background of less than 0.01 dru below 1 keV, as estimated by Geant4 [24] simulations.

The projected sensitivity of ECuME and DarkSPHERE in a DM-nucleon spin-independent cross-section versus DM mass is shown in Figure 3. A publication is under preparation which will detail the wider physics potential of DarkSPHERE, including spin-dependent nucleon interactions, DM-electron interactions [25], and other rare-event searches.

## 3  Summary

NEWS-G are using spherical proportional counters to shed light on unexplored DM candidates in the 0.05−10 GeV DM mass region. The current NEWS-G detector is yielding first results, and is currently taking data for its first physics campaign in SNOLAB. In parallel, the collaboration is working towards its future detectors, which will employ intact, underground electroformed detectors to suppress radioactive contamination in the detector material. A prototype for this, miniECuME, will begin construction in late 2022, with the full-scale detector ECuME, and the planned larger-scale detector, DarkSPHERE, following in the future.

## Acknowledgements

**Funding information**    This project has received support from UK Research and Innovation Science and Technology Facilities Council through grants No. ST/S000860/1, ST/V006339/1, and ST/W005611/1.

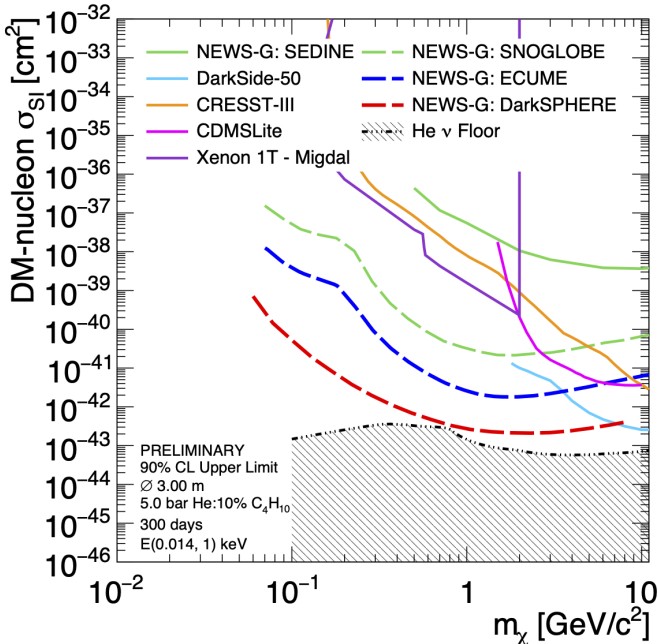

Figure 3: Projected sensitivity of current [10] and future NEWS-G detectors in DM-nucleon spin-independent cross section versus DM mass compared to current results from CDMSLite [26], CRESST-III [27], DarkSide-50 [28], and Xenon 1T (Migdal) [29] .

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
