# Peer review of "Status and Future Prospects of the NEWS-G Experiment"

_SciPost Physics Proceedings, doi:SciPost Phys. Proc. 12, 024 (2023)_

## Round 1 · Referee Report · Anonymous (Referee 1) · 2022-10-19

Report

In this proceeding authors describing search for light dark matter using spherical proportional counters. The manuscript is clearly written and well organised. It is also suitably formatted for publication. I recommend the manuscript for publication with some recommendations.

Requested changes

I recommend the manuscript for publication with some recommendations: - “Access to this region of parameter space requires either novel detection channels [5] or lower energy thresholds and light targets.” Can you provide references for “lower energy thresholds and light targets”? - “This produced the most stringent constraint on the spin-indpendent DM-nucleon scattering cross-section, 4.4 × 10−37 cm2, for a 0.5 GeV mass DM candidate [10] ” misleading - result was published in 2017 and not anymore the most stringent. Rephrase or clarify? - “The detector has since been installed inside…” - heavy, rephrase please - “The next generation NEWS-G detectors will be constructed using ultra-pure copper electroforming [?] ” fix the referring - On Fig1. Logo would be good to enlarge - “The projected sensitivity of ECuME and DarkSPHERE in a DM-nucleon spin-independent cross-section versus DM mass is shown in Figure ??.” - provide correct link to the figure - “Geant4 simulations” - provide references for Geant4 - Provide references to experiments shown at Fig3. - Please check Acknowledgements if they are correct. - Fix ref 14.

---

## Round 2 · Author Response

Dear Editors,
The authors would like to thank the reviewer for their thorough reading of the manuscript and for their helpful comments, which have improved its clarity in this resubmission. We have addressed each of their comments directly in the list of changes.
Best Regards,
Patrick Knights, on behalf of the authors.

---

## Round 2 · List of Changes

Changes were made in response to the reviewer's comments.

- “Access to this region of parameter space requires either novel detection channels [5] or lower energy thresholds and light targets.” Can you provide references for “lower energy thresholds and light targets”?
>>We have added a reference to the recent APPEC committee report [2] at this point.

- “This produced the most stringent constraint on the spin-independent DM-nucleon scattering cross-section, 4.4 × 10−37 cm2, for a 0.5 GeV mass DM candidate [10] ” misleading - result was published in 2017 and not anymore the most stringent. Rephrase or clarify?
>>Thank you for highlighting this, we have rephrased it to more accurately convey that it was the most stringent constraint at that time.

- “The detector has since been installed inside…” - heavy, rephrase please
>>We have rephrased this to improve clarity.

- “The next generation NEWS-G detectors will be constructed using ultra-pure copper electroforming [?] ” fix the referring
>>Fixed.

- On Fig1. Logo would be good to enlarge
>>While we do not disagree with you, this figure is a standard one that we would prefer not to change at this point.

- “The projected sensitivity of ECuME and DarkSPHERE in a DM-nucleon spin-independent cross-section versus DM mass is shown in Figure ??.” - provide correct link to the figure
>>Fixed.

- “Geant4 simulations” - provide references for Geant4
>>Fixed.

- Provide references to experiments shown at Fig3.
>>We have added the references.

- Please check Acknowledgements if they are correct.
>>Thank you for highlighting this oversight, this has now been amended.

- Fix ref 14.
>>Thank you for pointing this out. We would like this to reference the proceedings of this conference, 'Search for light WIMP recoils on methane with NEWS-G', by F. A. Vazquez de Sola Fernandez, on behalf of the NEWS-G collaboration. We hope that this can be achieved at the proofs stage.

You are currently on this page

Resubmission scipost_202210_00039v2 on 25 November 2022

---

## Editorial Decision

published